# Associations of Adolescents’ Excessive Electronic Device Use, Emotional Symptoms, Sleep Difficulty, and Communication with Parents: Two-wave Comparison in the Czech Republic

**DOI:** 10.3390/children9081186

**Published:** 2022-08-08

**Authors:** Yi Huang, Jinjin Lu

**Affiliations:** 1Department of Psychology, Faculty of Social Studies, Masaryk University, 602 00 Brno, Czech Republic; 2Institute for Research of Children, Youth and Family, Faculty of Social Studies, Masaryk University, 602 00 Brno, Czech Republic; 3Department of Educational Studies, Xi’an Jiaotong-Liverpool University, Suzhou 215123, China

**Keywords:** excessive electronic device use, psychological symptoms, parent–adolescent communication, HBSC

## Abstract

Adolescents’ excessive electronic device use is associated with psychological problems. However, it is unknown which psychological symptom, including emotional symptoms and sleep difficulty, correlates with excessive electronic device use most strongly. Besides, according to the social displacement theory, parent–adolescent communication might mediate the relationship between excessive electronic device use and psychological symptoms. Using the Czech national survey Health Behaviour in School-aged Children (HBSC) data in the years 2006 (*n* = 4782) and 2014 (*n* = 5082), we used network analysis to explore the relationship between psychological symptoms and excessive electronic device use. In addition, we conducted a mediation analysis to examine the role of parent–adolescent communication. The results revealed that excessive electronic device use correlated most strongly with adolescents’ irritability or bad temper, and this conclusion was stable in 2006 and 2014. In 2014, parent–adolescent communication mediated the relationship between adolescents’ excessive electronic device use and their psychological symptoms. The findings suggest that as the internet industry grows, it is essential to improve parent–adolescent communication quality to prevent adolescents’ psychological problems caused by excessive electronic device use.

## 1. Introduction

Adolescents’ excessive electronic device use has become a global public health problem as the internet industry accelerates. According to a UK survey, adolescents’ online time in 2015 more than doubled that of 2005 [1]. An extensive national study in Japan in 2020 found that adolescents averagely spend 1.81, 0.99, and 0.74 h on smartphones, social media, and online gaming, respectively. In addition, time spent on these internet-related activities had increased significantly compared to 2019 [2]. A study based in seven European countries noted that the trend of adolescents’ excessive internet use has increased [3]. Moreover, with easier access to the Internet via smartphones, adolescents have spent more time on online games and social networking sites [3].

It is worth noting that excessive electronic-device-based activities, such as problematic social media use and online gaming, are potential risk factors for adolescents’ psychological development. A narrative review showed consistent evidence that adolescents’ excessive social media use may lead to adverse psychosomatic conditions (e.g., sleep difficulties), psychological problems (e.g., anxiety and depression), and social developmental deficits (e.g., hostility and loneliness) [4]. Several meta-analysis studies also certified the relationship between children’s and adolescents’ excessive social media use and emotional disorders [5,6,7]. In addition, adolescents diagnosed with internet gaming disorder have more sleep problems [8]. A cross-sectional study noted that excessive internet use fits a psychopathological model, which means that it is a health risk behaviour that may lead to psychopathological symptoms such as emotionality, social-interactions problems, and conduct problems [9].

Parent–adolescent interactions may mediate the relationship between adolescents’ excessive electronic device use and psychological symptoms. The social displacement theory argues that adolescents’ excessive time in the virtual world may reduce their real-life interactions with parents, and consequently increase the risk of psychiatric symptoms [10]. The social displacement theory initially marked the effect of social media on interpersonal relationships. It suggested that time spent on social media replaces the time spent in face-to-face interaction with family and friends, which is highly correlated with individuals’ wellbeing. Before the personal computer (PC) era, TV was the main contributor to social displacement. In the PC era and current post-PC era, the Internet and social media are seen as the decisive reasons for social displacement, respectively [11,12]. The theory by Subrahmanyam et al. also suggested that excessive online time causes parent–adolescent conflicts in the offline world that result in adolescents’ psychological health problems [13]. In fact, it was found that adolescents’ intensive social media use is indirectly associated with depressive symptoms through perceived parental support, which meant that during the parent–adolescent interaction, adolescents’ subjective feelings of parental support mediate the relationship between adolescents’ excessive social media use and their depression [14]. Similarly, another study discussed the mediator role of parent–adolescent communication. In detail, adolescents’ internet gaming addiction leads to a poor quality of father–adolescent communication, eventually increasing the risk of adolescents’ aggressive behaviours [15]. Therefore, the underlying mechanism of the relationship between adolescents’ excessive electronic device use and their psychological health might be the mediation effect of parent–adolescent interactions. This is especially so since stepping into the post-PC (personal computer) era, which started in the late 2000s and early 2010s, with the trend of adolescents’ daily social media usage through the smartphone increasing significantly [16,17]. As discussed above, adolescents’ excessive communication through social media may decrease parent–adolescent communication, increase parent–adolescent conflicts, and exacerbate psychological health problems. Thus, the quality of parental communication has become increasingly important due to the social contextual changes.

Consistent with the global trend, the time that Czech adolescents use electronic devices has increased for both genders [18]. In 2006, the proportion of 15-year-old girls who spent less than 2 h per day on electronic devices was 68.7%, while in 2010 and 2014, the proportions were 42.3% and 36.8%, respectively. Likewise, among 15-year-old boys, the proportion who spend within the recommended time limitation on their electronic devices, which is no more than 2 h daily [19], has decreased significantly.

Despite a few reviews indicating that excessive electronic-device-based activities may lead to psychological symptoms such as depression and sleep difficulty [20,21], to our best knowledge, not a single study comprehensively compared the correlative strengths of relationships between excessive electronic device use and psychological problems. Such research is also lacking in the Czech Republic. Furthermore, the trend of the correlations remains unknown in Czechia. Additionally, the social displacement theory underscored the importance of a good parent–adolescent relationship for adolescents’ psychological health in the current electronic era. However, the possible positive parental role in the relationship between excessive electronic device use and adverse psychological outcomes among Czech adolescents has not previously been examined.

Generally, psychological symptoms include three emotional symptoms, which are depression, anxiety, and irritability, and a symptom of sleep difficulty [22]. The current study aimed to compare the strengths of associations between each psychological symptom and adolescents’ excessive electronic device use, and determine the symptom connected to excessive electronic device use most strongly in 2006 and 2014. The strongest symptom associated with excessive electronic device use would be considered the most vital symptom connected to excessive electronic device use. We aimed to investigate if the most critical symptom linked to excessive electronic device use remained the same across the two waves (2006-year wave and 2014-year wave). In addition, this research aimed to examine if adolescents’ communication with their parents can mediate the association between excessive electronic device use and their emotional symptoms and sleep difficulty.

## 2. Materials and Methods

### 2.1. Data

This study adopted the data of the Health Behaviour in School-aged Children (HBSC) survey conducted in 2006 and 2014 in the Czech Republic. HBSC is a cross-national project launched by the World Health Organization and aims to investigate adolescents’ health conditions, health behaviours, wellbeing, and their background information. Two-stage sampling procedures were followed in each participating country to obtain a representative national sample. In the first stage, schools were chosen randomly, while in the second stage, adolescents at 11/13/15-year-old specific national grades participated in the survey voluntarily and anonymously. The ethical approval for the data collection was granted by the Ethical Committee of the Faculty of Physical Culture, Palacký University in Olomouc, Czech Republic, No.17/2013, on 25 March 2013. There were 4782 eligible observations in the 2006-year sample and 5082 in the 2014-year sample.

### 2.2. Measurements

#### 2.2.1. Psychological Symptoms

We used the psychological subscale of the HBSC Symptom Checklist to measure three emotional symptoms of adolescents (feeling low, feeling nervous, and irritability or bad temper) and sleep difficulty. Participants were required to rate the frequency of experienced symptoms in the last six months from 1 (“about every day”) to 5 (“rarely or never”). I reversed the scores. Thus, a higher score for each item referred to a more frequently experienced symptom. The reliability of the scale was acceptable in 2006 (Cronbach’s Alpha = 0.73) and 2014 (Cronbach’s Alpha = 0.76). 

#### 2.2.2. Excessive Electronic Device Use

We measured the time in hours spent using electronic device on weekdays and weekends. In 2006, participants responded to the item “About how many hours a day do you usually spend using a computer for chatting online, browsing the Internet, emailing, doing homework, etc., in your free time for weekdays and weekend?”. In 2014, the item changed to “How many hours a day, in your free time, do you usually spend using electronic devices such as computers, tablets (such as iPad) or smartphones for other purposes, for example, homework, emailing, tweeting, Facebook, chatting, surfing the Internet for weekdays and weekend?”. The response ranged from 1 (“never”) to 9 (“7 h or more”). Based on the previous experience [23], we computed the average electronic device use time on weekdays and weekends to obtain a single measurement of electronic device use time. According to the American Academy of Pediatrics [19], screen time over 2 h daily is not encouraged. Therefore, we categorised the participants spending less than 2 h daily on electronic device as the “non-excessive electronic device use” group and those spending at least 2 h daily on electronic device as the “excessive electronic device use” group. 

#### 2.2.3. Communication with Parents

Adolescents answered two questions: “How easy is it for you to talk to your father about things that really bother you?” and “How easy is it for you to talk to your mother about things that really bother you?”. They responded from 1 (“very easy”), 2 (“easy”), 3 (“difficult”), 4 (“very difficult”), to 5 (“don’t have or see this person”). The supportive talks between parents and adolescents about daily problems are highly related to adolescents’ wellbeing [24]. The easy talk about life problems reflects good communication quality, while the difficult talk reflects poor communication quality. The communication quality is considered the worst if parents totally refuse to talk about their children’s life problems. We reversed and averaged the scores of the two items as a single index of communication with parents, which meant a high score indicated a better quality of communication with parents.

### 2.3. Data Analysis

To identify the most vital psychological symptom connected to adolescents’ excessive electronic device use in 2006 and 2014, I adopted the R package “bootnet” to conduct the bootstrap network analysis. A network graph visually demonstrates the correlation between every two variables, which helps us easier understand the complex relationship between variables. In a network structure, each node represents a variable, while each edge represents the partial correlation between two nodes while controlling the correlations to other left nodes [25]. Unlike the traditional hypothesis-driven analysis, for instance, regression analysis, network analysis is a typical data-driven approach. Thus, network analysis can also help researchers determine the most important variables linked to the target variable without any hypothesis. In the current study, all symptoms were in the first cluster, while the excessive electronic device use was a single node in the second cluster. By using the bridge strength centrality analysis, we explored the symptom most strongly connected to adolescents’ excessive electronic device use. Bridge strength centrality aims to sum the absolute values of edge weights extending from a node to all other nodes in another cluster in the network structure. The stability of the network framework was estimated by the correlation stability (CS). We adopt the case-drop bootstrap procedures to compute CS. CS refers to the maximum portion of dropped cases of the total sample when the estimated centrality can still correlate to the original network at the 0.7 effect size level. If the portion occupies over 50%, the stability of the network is considered good [26].

Subsequently, we built two mediation models to probe the role of communication with parents in the relationship between excessive electronic device use and psychological symptoms for the 2006-year wave and the 2014-year wave, respectively. The mediation analysis was conducted through the R package “lavaan”.

## 3. Results

### 3.1. Descriptive Statistics

Descriptive statistics are shown in Table 1. The rate of adolescents spending excessive time on electronic devices in 2014 (48.9%) was higher than that in 2006 (24.5%). The samples in 2006 and 2014 had a balance in sex and age.

### 3.2. Network Analysis

Table 2 demonstrated the bridge strength centrality of each symptom connected to excessive electronic device use. In this case, bridge strength centrality referred to the sum of the partial correlations extending from a psychological symptom to excessive electronic device use. Thus, the higher strength centrality of a symptom indicated a stronger correlation to excessive electronic device use. The results (see Table 2) suggested that in 2006, adolescents’ excessive electronic device use was most strongly associated with the emotional symptom “irritability or bad temper”. The network based on the sample in 2006 was stable as the CS was 75%. 

In the 2014-year wave, the results also suggested that the top symptom linked to adolescents’ excessive electronic device use was “irritability or bad temper” (see Table 2). The symptoms “feeling low” and “feeling nervous” were also relatively highly correlated with excessive electronic device use in this wave. The CS was 75%, which indicated good stability of the network.

The network graphs in two waves can be seen in Figure 1 and Figure 2. It is worth noting that the association between the symptom “feeling nervous” and excessive electronic device use increased significantly, from 0.20 in 2006 to 0.89 in 2014.

In summary, adolescents’ excessive electronic device use correlated most strongly to the psychological symptom “irritability or bad temper” in 2006 and 2014. When compared to 2006, the correlations between electronic device use and the other two emotional symptoms “feeling low” and “feeling nervous” increased significantly from 2006 to 2014.

### 3.3. Mediation Analysis 

According to the correlation analysis based on the 2006-year-wave sample, adolescents’ excessive electronic device use was not significantly associated with communication with parents (Spearman’s rho = −0.01, *p* = 0.62). Therefore, even without a formal mediation analysis, we could conclude that communication with parents did not play the role of a mediator between adolescents’ excessive electronic device use and psychological symptoms.

In the 2014-year wave, adolescents’ excessive electronic device use significantly correlated to communication with parents (Spearman’s rho = −0.08, *p* < 0.01) and psychological symptoms, including feeling low (Spearman’s rho = 0.09, *p* < 0.01), feeling nervous (Spearman’s rho = 0.10, *p* < 0.01), irritability or bad temper (Spearman’s rho = 0.11, *p* < 0.01), and sleep difficulty (Spearman’s rho = 0.07, *p* < 0.01). These correlation results allowed us to continue the mediation analysis.

The standardised mediation model (see Figure 3) suggested that adolescents’ excessive electronic device use was directly and significantly associated to some extent with sleep difficulty, feeling low, feeling nervous, and irritability or bad temper. Moreover, adolescents’ excessive electronic device use was also indirectly associated with the above four symptoms through communication with parents. Thus, in this case, the total effect of excessive electronic device use on three emotional symptoms and the sleep difficulty symptom can be split into two parts: the direct effect of excessive electronic device use itself on symptoms, and the indirect effect of excessive electronic device use on symptoms through the pathway “excessive electronic device use→communication with parents→four symptoms”. In the pathway “excessive electronic device use→communication with parents→sleep difficulty”, the indirect effect—in another word, the mediation effect of “communication with parents”—was 72.7% of the total effect. Correspondingly, the effects of mediator “communication with parents” were 13.8%, 19.2%, and 13.0%, respectively, in the relationships between excessive electronic device use and irritability or bad temper, feeling low, and feeling nervous.

## 4. Discussion

The primary purposes of this study were (1) to determine the psychological symptom most strongly related to adolescents’ excessive electronic device use and (2) to examine the mediation effect of communication with parents between excessive electronic device use and psychological symptoms in the years 2006 and 2014. Our findings show that adolescents’ irritability or bad temper is the most vital symptom linked to excessive electronic use. We also raise the consideration of the critical role of parent–adolescent communication.

The first important finding is that adolescents’ irritability or bad temper is strongly linked to their excessive electronic use; this conclusion remains stable across the 2006-year wave and the 2014-year wave. Our measurement of adolescents’ electronic device use was mainly based on online activities. Therefore, in our study, the excessive time spent on electronic devices reflected their problematic internet use. The result echoes the social/cognitive addiction theory that suggests adolescents’ problematic internet use is due to insufficient emotion regulation [27], which may cause aggressiveness, impulsiveness, risk-taking, and a lack of inhibition [28,29]. Seo et al. found that in Korea, adolescents who spend excessive time online show more irritability, sexual avoidance, and criticism to others [30]. Similarly, among Turkish teenagers, compared to those without internet addiction, internet addicts are more anxious and irritable [31].

Interestingly, the findings suggest that the tendency of the associations between adolescents’ excessive electronic device use and the other two emotional symptoms (feeling low and feeling nervous) has become more intensive from 2006 to 2014. Even in this case, it was found that in 2014, the association between excessive electronic device use and the two emotional symptoms (feeling low and feeling nervous) were close to that between excessive electronic device use and irritability. The excessive social media use in the post-PC era can explain the phenomenon. According to the uses and gratifications theory, adolescents’ excessive social media use may reflect interpersonal rejection in reality [32]. To satisfy their self-promotion, adolescents facing interpersonal rejection tend to spend more time online to seek reassurance. It is known that in the long term, interpersonal rejection is associated with adolescents’ depressive and anxious emotions [33]. In turn, using social media excessively and escaping from reality strengthens their problematic interpersonal behaviours in reality, such as social withdrawal, which worsens their depressive and anxious symptoms, including feeling low and feeling nervous [34].

In the 2014-year wave, the results suggest adolescents’ communication with their parents can mediate the relationship between excessive electronic device use and four psychological symptoms: feeling low, feeling nervous, irritability or bad temper, and sleep difficulty. This finding is consistent with the social displacement theory, which indicates adolescents’ excessive time online decreases their communication with parents in the offline world and increases their conflicts with parents [10]. Empirically, it was found that adolescents who are addicted to the virtual world tend to spend more time alone and have a poorer quality of interpersonal relationships [30,35]. Parents play an essential role for adolescents during a critical period for developing a positive self-concept and self-identity. Thus, parent–adolescent interaction is highly correlated with adolescents’ mental health. Consequently, it is plausible that the current study suggests that adolescents’ excessive electronic use results in poorer communication with parents and eventually leads to worse psychological health. The study also found that, among the four symptoms, the mediation effect of parent–adolescent communication was strongest with sleep difficulty. It may be explained by the social media addicts’ screen usage time habit. A report from the American Academy of Pediatrics suggested social media addict adolescents tend to use social media at night and delay their sleep time, which is one of the main contributors to sleep difficulties [36]. Good parent–adolescent communication reduces adolescent online communication seeking. Thus, communication with parents significantly lowers adolescents’ risk of sleep difficulty.

However, in the 2006-year wave, we did not find an association between excessive electronic device use and communication with parents. The reason might be that, in 2006, adolescents mainly used electronic devices for purposes beyond social media, such as searching for homework information, reading, and emailing, rather than connecting to social media. As discussed in the Introduction, before the 2010s, we saw the PC era. Since the late 2000s and early 2010s, we have seen the evolution of the post-PC era, in which adolescents’ social media usage through smart electronic devices has increased significantly [16,17]. Other activities beyond social media usage rarely decreased the offline time with parents. Salehan and Negahban said that with the enormous growth of smartphone use, compared to the computer-based era, there has been a significant increase in the use of social networking services, especially among the youth [37]. Thus, in 2006, when social media was less prevalent, adolescents’ electronic device usage not aiming for social media did not decrease their offline communications with parents in real life, and it did not increase their conflicts with parents. Indeed, empirical evidence has proven that social benefit from the Internet increases parent–adolescent conflicts and communication difficulties [38]. In conclusion, it is reasonable to suggest that when social media is less widely used, electronic device use does not decrease parent–adolescent communication and family intimacy.

Our results underscore the need for intervention to improve parent–adolescent communication because it should help prevent emotional symptoms and the sleep difficulty symptom caused by adolescents’ excessive electronic device use. In fact, many researchers have advocated parental cooperation to relegate the adverse effects of the Internet on adolescents [39,40]. Moreover, according to previous studies, good parent–adolescent communication quality can decrease the risk of adolescents’ problematic internet use [41,42]. Considering the positive effects of communication with parents on reducing adolescents’ excessive internet use and the resulting negative outcomes, it is necessary to design related intervention programmes to improve parent–adolescent communication quality. There are three tips for parents. First, parents need to communicate with their children frequently. Parents should especially encourage children to talk about problems that they encounter. Supportive talk about daily problems helps adolescents develop coping strategies and positive self-identity, which decreases the possibility of adolescents’ excessive social media use to escape from reality or seek reassurance online. Second, it is necessary for parents to avoid accelerating parent–adolescent conflict. Adolescents’ excessive electronic device use may increase parent–adolescent conflict. On the one hand, conflict-filled communication leads to adolescent psychological symptoms. On the other hand, it may hinder adolescents’ attempts to further communicate with their parents. Third, it is a good idea for parents to discuss the excessive electronic device usage problem with their children. Such an open conversation helps discover the deep reason behind excessive electronic device use, for instance, bullying and academic difficulties. It also helps develop coping strategies for these problems.

We acknowledge there were several limitations in this study. First, we adopted cross-sectional data in two waves when building the mediation models. Due to the open-access database’s constraints and the HBSC protocol’s design, we could not track adolescents longitudinally. Therefore, we could not make cause–effect statements in the current study. We recommend that in future, researchers investigate the relationship between these variables using a longitudinal analysis. Furthermore, a dynamic-data-based design is highly recommended because dynamic models emphasise the change of variables in a development process, which helps us understand the cause–effect and reciprocal relationships between excessive electronic device use, parent–adolescent communication, and psychological symptoms [43]. For instance, as we discussed, there might be a longitudinal two-direction relationship between adolescents’ excessive electronic device use and parent–adolescent communication. If adolescents spend excessive time online, they may have less communication with their parents and, in turn, the lack of parent–adolescent communication may exacerbate adolescents’ smart device addiction. Second, the responses might be biased because all of the questionnaires were answered by adolescents. It is worth noting that the communication quality might differ from the parental perspective. Thus, it will be invaluable to conduct surveys of the parent population in the future. Third, except for adolescents living with two biological parents, the current study also included adolescents who were in single-parent families or reconstituted families during the survey time. It is plausible that compared to adolescents from two-parent families, adolescents from single-parent families or reconstituted families might have less communication with the parent who did not live with them. Therefore, in future, it is worth focusing on the role of communications with the primary caregiving parent. Moreover, it is necessary to compare the importance of communication with parents across two types of families. Fourth, we did not investigate if the mediation effect of communication with parents varies across sex and age. Regarding the variable “age”, it would be especially useful to investigate its effect by tracking adolescents longitudinally.

## 5. Conclusions

The network analysis suggests that among the psychological symptoms, adolescents’ irritability or bad temper is the most vital symptom correlated with their excessive electronic device use. In addition, the conclusion is stable across the years. The mediation analysis suggests that in the post-PC era, when access to the Internet has become much easier via smart devices, parent–adolescent communication mediates the relationship between adolescents’ excessive electronic device use and their emotional symptoms and sleep difficulty. The findings mark the important role of parent–adolescent communication in the central European region, because this helps prevent emotional symptoms and sleep difficulty caused by excessive electronic device use. Therefore, we advocate for intervention programmes to improve parent–adolescent communication quality.

## Figures and Tables

**Figure 1 children-09-01186-f001:**
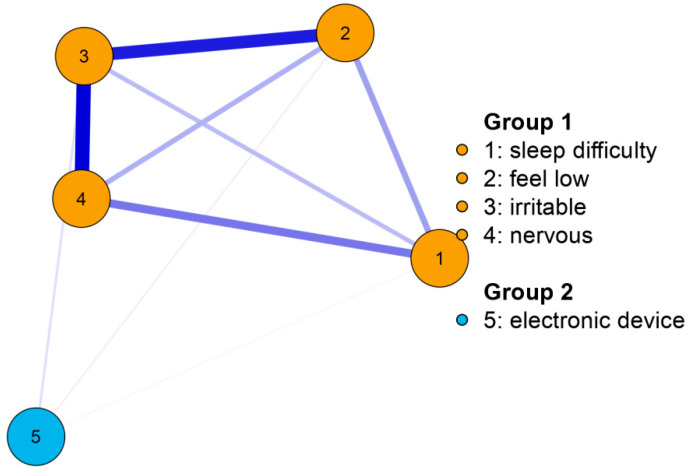
The network graph demonstrating the correlation between each psychological symptom and adolescents’ excessive electronic device use based on the 2006-year sample.

**Figure 2 children-09-01186-f002:**
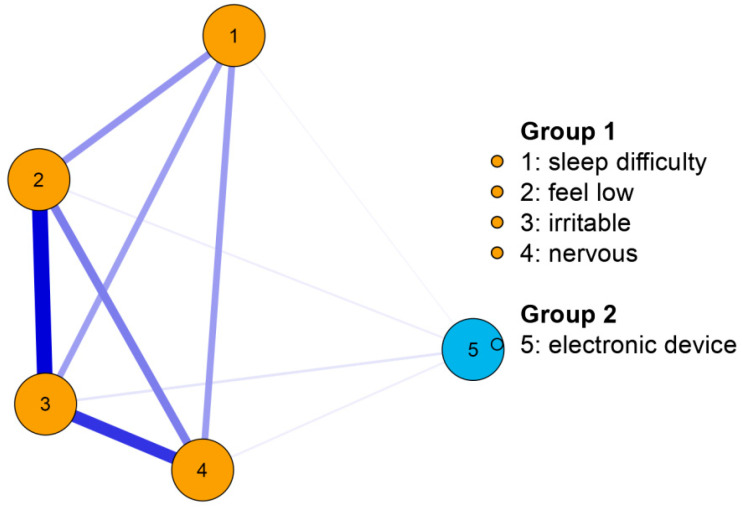
The network graph demonstrating the correlation between each psychological symptom and adolescents’ excessive electronic device use based on the 2014-year sample.

**Figure 3 children-09-01186-f003:**
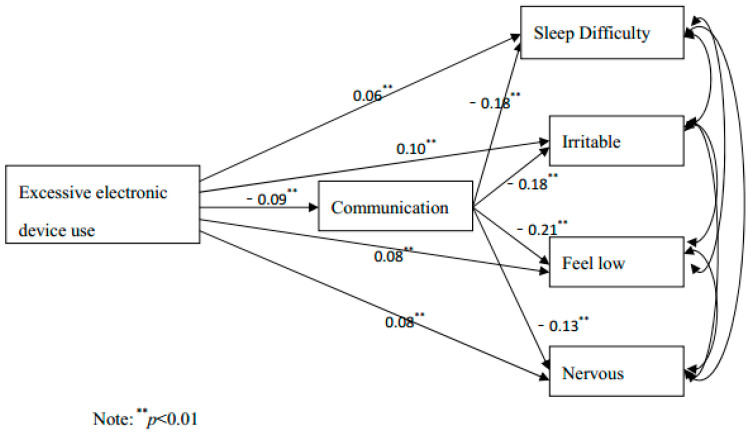
The mediation model showing the mediator role of parent–adolescent communication between excessive electronic device use and each psychological symptom.

**Table 1 children-09-01186-t001:** Descriptive statistics depicting the sample characteristics in two waves.

		*n* (%)	Mean	Std. Dev
**2006-Year Wave**				
**Sex**	Boy	2416 (50.5%)		
	Girl	2366 (49.5%)		
**Grade**	11-year-old grade	1509 (31.6%)		
	13-year-old grade	1601 (33.5%)		
	15-year-old grade	1665 (34.9%)		
**Symptoms**	Feeling low		1.94	1.16
1 (“about every day”) to 5 (“rarely or never”)	Feeling nervous		2.84	1.24
	Irritability or bad temper		2.73	1.14
	Sleep difficulty		2.09	1.34
**Electronic Device Use**	Not excessive	3572 (74.5%)		
	Excessive	1157 (24.5%)		
**Communication with Parents** 1 (“very poor”) to 5 (“very well”)	Communication with parents		3.72	0.89
**2014-Year Wave**				
**Sex**	Boy	2420 (47.6%)		
	Girl	2662 (52.4%)		
**Grade**	11-year-old grade	1596 (31.4%)		
	13-year-old grade	1749 (34.4%)		
	15-year-old grade	1737 (34.2%)		
**Symptoms**	Feeling low		2.048	1.274
1 (“about every day”) to 5 (“rarely or never”)	Feeling nervous		2.679	1.362
	Irritability or bad temper		2.62	1.251
	Sleep difficulty		2.134	1.44
**Electronic Device Use**	Not excessive	2528 (51.1%)		
	Excessive	2417 (48.9%)		
**Communication with Parents** 1 (“very poor”) to 5 (“very well”)	Communication with parents		3.87	0.90

**Table 2 children-09-01186-t002:** The strength centrality analysis demonstrating the link between adolescents’ excessive electronic device use and emotional symptoms and sleep difficulty symptom.

	Sleep Difficulty	Feeling Low	Irritable	Nervous
2006-year wave	0.15	0.59	0.96	0.20
2014-year wave	0.25	0.87	0.96	0.89

## Data Availability

HBSC data can be downloaded through the website http://www.hbsc.org/ (accessed on 30 May 2022).

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
