# Peer review of "Associations of Adolescents’ Excessive Electronic Device Use, Emotional Symptoms, Sleep Difficulty, and Communication with Parents: Two-wave Comparison in the Czech Republic"

_children, 2022, doi:10.3390/children9081186_

Round 1

Reviewer 1 Report

The manuscript has two aims which are to investigate whether electronic media device use is associated with emotional symptoms and sleep difficulty and to investigate whether communication with parents mediate the above associations. This is two interesting research questions. Y

However, there are some inconsistencies throughout the manuscript. First of all, the aim is actually not completely clear to me. You write that: the current study aimed to examine the associations of emotional symptoms, sleep difficulty symptom, and excessive electronic device use among Czech adolescents in the year 2006 and the year 2014. We aimed to compare if the most vital symptom connected to excessive electronic device use stayed the same crossing the two waves.”

Do you have two aims here? To examine associations AND to compare whether the variable with the strongest association with excessive electronic device were the same in 2006 and 2014? The author should make that clear in the text.

Another potential limitation is that the 11-, 13- og 15-year-olds are not investigated separately. What is the reason for that? From my view that would be more interesting than comparing 2006 and 2014. Psychological symptoms and electronic medica use are probably increasing across the age of 11 to 15 years and the communication with parents be even more important in the associations.

Generel comments:

The manuscript would certainly benefit from English proofreading. For example, line 66: “As the same as the global trend, the time of Czech adolescents' electronic device use has increased in both genders [14].

Moreover, the author should check for grammatical errors throughout their manuscript. For example, line 19: please replace ’irritability of ‘ with ‘irritability or‘.

Introduction:

In the first part of the introduction, one of the major references is rather old, i.e., adolescents from Singapore in 2007. Updated knowledge is needed to understand the problem. Or the following statement that “A study based on seven European countries noted that the trend of adolescents' excessive internet use has increased” should be more explicit. What is the prevalence today?  

Line 67: “In the year 2006, the rate of girls at 15 years old who spent less than 2 hours per day on electronic devices was 68.7%, and in 2010 and 2014, the rates were 42.3% and 36.8%, respectively”. It is not rates but prevalences or proportions.

Line 69: “the portion of individuals spending”: It is not portion but proportion

Line 70: “within the recommended time limitation, which is no more than two hours daily”: A reference is lacking

Line 73: “However, the correlation between adolescents' excessive electronic device use and the negative psychological symptoms remained unclear in the Czech Republic”: Is it only in the Czech Republic? I would like to get a deeper understanding of the international relevance of this study. Likewise: Line 75 “the psycho pathological mechanism of the possible relationship between excessive electronic device use and negative psychological outcomes among Czech adolescents was not discussed previously”: Has it been discussed (or rather investigated) internationally? And what do you mean with “psycho-pathological mechanism” in relation to this study?

Materials and Methods:

Generally, in this section, the author uses “we”; however, it seems that there is only one author?

The author uses data from 2006 and 2014. Electronic media use is rapidly evolving and therefore, data is quickly getting outdated. To my knowledge, HBSC data is collected with about four years interval. Therefore, I am wondering why newer data not is used?

Line 107: Something is lacking in “about every”. Maybe day?

Line 113: We measured the time of electronic device use daily on school days and weekends”. I think the author should rewrite the sentence to e.g.: “the span of time during which electronic device was used for school days and weekends” or hours spent using electronic device for school days and weekends

Line 114: “Participants reported the time of electronic device use (e.g., computers, tablets, or smart phones) for other purposes, for example, homework, online chatting, internet and emailing, and social media, like Facebook and Twitter.” What do you mean with “other purposes? You need to include a “than” for the sentence to make sense. Moreover, instead of writing “the time of electronic device use” it would be more correct to write e.g., “the frequency of electronic device use”. This comment is also related to my comment above.

Line 129: We reversed and averaged the scores of the two items as a single index of communication with parents, which meant a high score indicated a better quality of communication with parents”: I would be nice to have a discussion of this scale – what is the limitation and strength of including the response of those who don’t see or have the parent; of summarising on both parents and not keeping it to each parents.

As a reader it can be difficult to follow the statistical analysis. There is a very short description on page 3. Particularly the network analysis should be explained more explicitly and there should be a reference for this method. This is a major limitation which need to be addressed.

Results:

Table 1: Please insert the range of the scales.

Table 1: Communication with parents: Please insert the direction (e.g. “Good”)

Page 5: The title of section 3.2, “Network analysis”, is confusing because it can also mean the analysis of e.g. the network of social media users. As mentioned above, you need to describe this method in more detail and why it is the best available method; however, this should be done in the Data analysis section. Therefore, the following information should be placed in the data analysis section:Besides, we estimated the network correlation-stability (CS) by the method of case-drop bootstrap. CS refers to the maximum portion of dropped cases of the total sample when the estimated centrality can  still correlate to the original network at the 0.7 effect size level. If the portion occupies over 50%, the stability of the network is considered good [19].”

The text explaining table 2 should be more informative. Because the statistical method is not a commonly used method, the author needs to explain the results from the network analysis in more details or at least report on the results rather than only referring to table 2. How should 0.96 be interpreted? I am also wondering why you’re not commenting on the development in feeling nervous. The development from 0.20 in 2006 to 0.89 in 2014 seems remarkable. Furthermore, the associations of feeling low and feeling nervous seem close to the association of feeling irritable in 2014 – or how do you interpret this?

Figure 1 and figure 2 are completely identical. Is this a mistake? From table 2, the correlations are not the same.

Line 206: “the mediation effects of communication with parents were 72.7%, 13.8%, 19.2%, and 13.0%, respectively, in the relationships between excessive electronic device use and four symptoms, sleep difficulty, irritability or bad temper, feeling low, and feeling nervous”. How should this sentence be interpreted? That parent-child communication explains 73% of the association between electronic media use and poor sleep, and 14% of the association with feeling irritable? These results would be very interesting, however, the author does not comment on these results in the Discussion.

Discussion:

Line 235: “This finding consists of the social displacement theory”: I am not sure what the author mean with the finding ‘consist of’? The finding can be explained by a theory.

There is no discussion of how much parent-child communication mediated the examined associations although this is reported in the results.

There is no discussion of why confounders are not included. I guess this could be a bias in this study.

Line 274: “making the cause-effect statement less strict in the current study”. What do the author mean with ‘less strict’? Cross sectional data cannot conclude a cause and effect relationships, however, it would be nice to include a discussion of whether there could be reverse causality and potential confounders.

Line 278: It is stated that a dynamic-data-based design is recommended. However, it is unclear to me what this is.

Conclusion: The conclusion is not completely answering the aims. It could be improved by being more concise.

Author Response

Dear reviewer, thank you very much for your valuable comments and suggestions! Based on all your comments, we would like to respond. Please kindly find the responses below. Thanks again.

Comment 1. The manuscript has two aims which are to investigate whether electronic media device use is associated with emotional symptoms and sleep difficulty and to investigate whether communication with parents mediate the above associations. This is two interesting research questions.

However, there are some inconsistencies throughout the manuscript. First of all, the aim is actually not completely clear to me. You write that: “the current study aimed to examine the associations of emotional symptoms, sleep difficulty symptom, and excessive electronic device use among Czech adolescents in the year 2006 and the year 2014. We aimed to compare if the most vital symptom connected to excessive electronic device use stayed the same crossing the two waves.”

Do you have two aims here? To examine associations AND to compare whether the variable with the strongest association with excessive electronic device were the same in 2006 and 2014? The author should make that clear in the text.

Response 1. Dear reviewer, thank you very much for your valuable comment and we would like to respond to it. The main aim is to find out the most vital symptom connected to excessive electronic device use, based on the correlations between psychological symptoms and excessive electronic device use, which means, in the beginning, it is necessary to examine the associations of these symptoms and excessive electronic use. And it is the reason why we adopted network analysis: network analysis is based on the calculation of partial correlative effect size between every two variables while controlling the correlations between the two variables and other variables. So, we clarified the study aim in more detail according to the above discussions. Please kindly find the revisions in the last paragraph of "introduction". We also stated why we used network analysis by adding the above discussions in the section "data analysis". Please kindly find them. Thank you in advance.

Comment 2. Another potential limitation is that the 11-, 13- og 15-year-olds are not investigated separately. What is the reason for that? From my view that would be more interesting than comparing 2006 and 2014. Psychological symptoms and electronic medica use are probably increasing across the age of 11 to 15 years and the communication with parents be even more important in the associations.

Response 2. Thank you for your comment very much. Please allow us to explain here, as well as in the main text of the manuscript why we compared the year waves, instead of the age waves. Thank you. In fact, compared to the year 2006, which was a Personal Computer (PC) based era. But after stepping into the post-PC era (after the early 2010s), electronic devise use is more and more linked to social media use through smartphones. Therefore, adolescents face a decrease in offline communication with parents because they spend more time on online communicating with others than previously. Thus, due to the social contextual change, it is interesting and valuable to investigate the changes in parental role across years in the relationship between adolescents' excessive electronic device use and psychological health. We added the information in the third paragraph in the "introduction" section. The revisions were highlighted in yellow. Please kindly find them. Thanks a lot.

Comment 3. Generel comments:

The manuscript would certainly benefit from English proofreading. For example, line 66: “As the same as the global trend, the time of Czech adolescents' electronic device use has increased in both genders [14].

Moreover, the author should check for grammatical errors throughout their manuscript. For example, line 19: please replace ’irritability of ‘ with ‘irritability or‘.

Response 3. Thank you for your suggestion. The new manuscript has received proofreading. The sentence "As the same as the global trend, the time of Czech adolescents' electronic device use has increased in both genders" was replaced with " Consistent with the global trend, the time that Czech adolescents use electronic device has increased for both genders ". And the " irritability of " was replaced by "irritability or" in the abstract section.

Comment 4. Introduction:

In the first part of the introduction, one of the major references is rather old, i.e., adolescents from Singapore in 2007. Updated knowledge is needed to understand the problem. Or the following statement that “A study based on seven European countries noted that the trend of adolescents' excessive internet use has increased” should be more explicit. What is the prevalence today? 

Response 4. Thank you for your suggestions. We removed the old Singaporean study and updated by adding a newer Japanese study to depict the adolescents' electronic device use situation. Please kindly find the highlighted text in the first paragraph of the "introduction". Regarding the cited article about "seven European countries noted that the trend of adolescents' excessive internet use has increased", we are sorry we cannot explicit the recent prevalence , because in this study, a screening instrument Excessive Internet Use Scale was used, which is not a diagnostic instrument. Thus, it is not possible to extract information about the prevalence of excessive internet use from this article.

Comment 5. Line 67: “In the year 2006, the rate of girls at 15 years old who spent less than 2 hours per day on electronic devices was 68.7%, and in 2010 and 2014, the rates were 42.3% and 36.8%, respectively”. It is not rates but prevalences or proportions. Line 69: “the portion of individuals spending”: It is not portion but proportion

Response 5. Thank you for pointing out the shortcoming. We replaced the "rate" and "rates" with "proportion" and "prevalences" in the revised manuscript. Also, we revised the word "portion" as "proportion".

Comment 6. Line 70: “within the recommended time limitation, which is no more than two hours daily”: A reference is lacking

Response 6. Thank you for pointing out this. We added the reference in the revised manuscript: American Academy of Pediatrics, Committee on Public Education. American Academy of Pediatrics: children, adolescents, and television. Pediatrics 2001;107:423–6.

Comment 7. Line 73: “However, the correlation between adolescents' excessive electronic device use and the negative psychological symptoms remained unclear in the Czech Republic”: Is it only in the Czech Republic? I would like to get a deeper understanding of the international relevance of this study.

Response 7. Thank you very much for your comment. Thus, in the new version, we wrote a single paragraph to elaborate on more details. Please find the 5th paragraph. First, we cited two literature review works that listed some typical psychological symptoms related to excessive electronic device use. Subsequently, we noted " to our best knowledge, not a single study comprehensively compared correlative strengths of relationships between excessive electronic device use and psychological problems. Such research is also lacking in the Czech Republic. Furthermore, the trend of the correlations remains unknown in Czechia.".

Comment 8. Likewise: Line 75 “the psycho pathological mechanism of the possible relationship between excessive electronic device use and negative psychological outcomes among Czech adolescents was not discussed previously”: Has it been discussed (or rather investigated) internationally? And what do you mean with “psycho-pathological mechanism” in relation to this study?

Response 8. Regarding to word "psychopathological mechanism", after cautious reckon, we would replace it with "parental role", because actually one of the main research purposes was to examine the mediator role of parent-adolescent communication in the relationship between excessive electronic device use and the negative psychological outcomes. Also, as written in the third paragraph in "introduction", we noted social displacement theory discussed parent-adolescent relationship should be a mediator in the relationship between excessive electronic device use and the negative psychological outcomes. Thus, in the new-version manuscript, we rewrote the sentences as " Additionally, the social displacement theory underscored the importance of a good parent-adolescent relationship for adolescents' psychological health in the current electronic era. Yet, the possible positive parental role in the relationship between excessive electronic device use and adverse psychological outcomes among Czech adolescents has not been examined previously.".

Comment 9. Materials and Methods:

Generally, in this section, the author uses “we”; however, it seems that there is only one author?

Response 9. Thank you very much for your question. In fact, there was another author participating in manuscript revision. So, in this new manuscript, there are two authors.

Comment 10. The author uses data from 2006 and 2014. Electronic media use is rapidly evolving and therefore, data is quickly getting outdated. To my knowledge, HBSC data is collected with about four years interval. Therefore, I am wondering why newer data not is used?

Response 10. Thank you for your question. Please allow us to explain it. Thank you. World Health Organization only opens access to the previous data to the public. The lasted data was collected in 2017, but WHO did not make this batch of data open yet. The latest accessible data was collected from 2013 to 2014.

Comment 11. Line 107: Something is lacking in “about every”. Maybe day?

Response 11. Thank you very much, Yes, after checking the original questionnaire, we revised the item response 1 as "about every day".

Comment 12. Line 113: “We measured the time of electronic device use daily on school days and weekends”. I think the author should rewrite the sentence to e.g.: “the span of time during which electronic device was used for school days and weekends” or hours spent using electronic device for school days and weekends

Response 12. Thank you. We rewrote this sentence as " We measured the time in hours spent using electronic device on weekdays and weekends.S".

Comment 13. Line 114: “Participants reported the time of electronic device use (e.g., computers, tablets, or smart phones) for other purposes, for example, homework, online chatting, internet and emailing, and social media, like Facebook and Twitter.” What do you mean with “other purposes? You need to include a “than” for the sentence to make sense. Moreover, instead of writing “the time of electronic device use” it would be more correct to write e.g., “the frequency of electronic device use”. This comment is also related to my comment above.

Response 13. Thank you for your question. In fact, the original item in the 2014 HBSC questionnaires was: " How many hours a day, in your free time, do you usually spend using electronic devices such as computers, tablets (like iPad) or smart phones for other purposes, for example, homework, emailing, tweeting, Facebook, chatting, surfing the internet? " "for other purpose" here means that using electronic device is not aiming to practice how to use electronic device. And in the new manuscript, we presented the original item in 20016 and 2014.

Comment 14. Line 129: We reversed and averaged the scores of the two items as a single index of communication with parents, which meant a high score indicated a better quality of communication with parents”: I would be nice to have a discussion of this scale – what is the limitation and strength of including the response of those who don’t see or have the parent; of summarising on both parents and not keeping it to each parents.

Response 14. Thank you very much for your suggestion. We agree it is important to discuss the limitation of the measurement of adolescent-parent communication. So I noted in the discussion in the new version that " Third, except for adolescents living with two biological parents, the current study also included adolescents who were in single-parent families or reconstituted families during the survey time. It is plausible that compared to adolescents from two-parent families, adolescents from single-parent families or reconstituted families might have less communication with one of the parents who did not live with them. Therefore, it is worth focusing on the role of communications with the primary caregiving parent in the future. Moreover, it is valuable to compare the importance of communication with parents across two types of families.". Please kindly find the added information. Thank you.

Comment 15. As a reader it can be difficult to follow the statistical analysis. There is a very short description on page 3. Particularly the network analysis should be explained more explicitly and there should be a reference for this method. This is a major limitation which need to be addressed.

Response 15: Thank you very much for your comment and suggestion. In the "data analysis" section, we elaborated on more details about the network analysis method. We also explained the step of correlation stability (CS) computation in this section. Please kindly find the new "data analysis" section.  

Comment 16. Results:

Table 1: Please insert the range of the scales.

Table 1: Communication with parents: Please insert the direction (e.g. “Good”)

Response 16. Thank you for your suggestion. Please kindly find Table 1 in the new manuscript. We inserted the range of scales and the direction of the measurement "communication with parents".

Comment 17. Page 5: The title of section 3.2, “Network analysis”, is confusing because it can also mean the analysis of e.g. the network of social media users. As mentioned above, you need to describe this method in more detail and why it is the best available method; however, this should be done in the Data analysis section. Therefore, the following information should be placed in the data analysis section: “Besides, we estimated the network correlation-stability (CS) by the method of case-drop bootstrap. CS refers to the maximum portion of dropped cases of the total sample when the estimated centrality can  still correlate to the original network at the 0.7 effect size level. If the portion occupies over 50%, the stability of the network is considered good [19].”

Response 17. Thank you for your comment. Yes, we agreed it is necessary to explain in detail about the network analysis method in "data analysis" section. Thus, as discussed in our "response 15", we added more details and described the advantages of network analysis method in the "data analysis" section. Also, we moved the CS computation to "data analysis" section.

Comment 18. The text explaining table 2 should be more informative. Because the statistical method is not a commonly used method, the author needs to explain the results from the network analysis in more details or at least report on the results rather than only referring to table 2. How should 0.96 be interpreted? I am also wondering why you’re not commenting on the development in feeling nervous. The development from 0.20 in 2006 to 0.89 in 2014 seems remarkable. Furthermore, the associations of feeling low and feeling nervous seem close to the association of feeling irritable in 2014 – or how do you interpret this?

Response 18. Thank you for your valuable comment very much! In the new revised version, first, we interpreted the meanings of numbers in Table 2. Second, we noted the development of symptoms "feeling nervous" and "feeling low". Third, we added the following information in results: " the symptoms "feeling low" and "feeling nervous" also relatively highly correlated to excessive electronic device use in the 2014-year wave. And, the correlations between electronic device use and other two emotional symptoms "feeling low" and "feeling nervous" aggravated significantly from 2006 to 2014." Moreover, we discussed the more intensive relationship between emotional symptoms and electronic device use in the "discussion" section. Please kindly find the third paragraph in "discussion" section.

Comment 19. Figure 1 and figure 2 are completely identical. Is this a mistake? From table 2, the correlations are not the same.

Response 19. Thanks for pointing out the mistake. We have already replaced Figure 2 with the correct figure. Please kindly check it.

Comment 20. Line 206: “the mediation effects of communication with parents were 72.7%, 13.8%, 19.2%, and 13.0%, respectively, in the relationships between excessive electronic device use and four symptoms, sleep difficulty, irritability or bad temper, feeling low, and feeling nervous”. How should this sentence be interpreted? That parent-child communication explains 73% of the association between electronic media use and poor sleep, and 14% of the association with feeling irritable? These results would be very interesting, however, the author does not comment on these results in the Discussion.

Response 20.  Thank you for your comment and suggestion. For explaining it more clearly. I re-wrote the sentences in the "results". Please kindly find the following sentences in the new manuscript: "Thus, in this case, the total effect of excessive electronic device use on three emotional symptoms and sleep difficulty symptom could break into two parts: the direct effect of excessive electronic device use itself on symptoms, and the indirect effect of excessive electronic device use on symptoms through the pathway "excessive electronic device use→communication with parents→four symptoms". In the pathway "excessive electronic device use→communication with parents→sleep difficulty", the indirect effect, in another word, the mediation effect of "communication with parents", was 72.7% of the total effect. Correspondingly, the effect of mediator "communication with parents" were 13.8%, 19.2%, and 13.0%, respectively, in the relationships between excessive electronic device use and irritability or bad temper, feeling low, and feeling nervous. "

And, we discussed the strongest mediate effect of parent-adolescent communication on sleep difficulty. Please find the following added information in the discussion section: " The study also found that, among the four symptoms, the mediation effect of parent-adolescent communication was strongest on sleep difficulty. It may be explained by the social media addicts' screen usage time habit. A report from the American Academy of Pediatrics suggested social media addict adolescents tend to use social media at night and delay their sleep time, which is one of the main contributors to sleep difficulties [36]. Good parent-adolescent communication reduces adolescent online communication seeking. Thus, communication with parents lowers adolescents' risk of sleep difficulty significantly.".

Comment 21. Discussion:

Line 235: “This finding consists of the social displacement theory”: I am not sure what the author mean with the finding ‘consist of’? The finding can be explained by a theory.

Response 21. Thank you for your comment. We re-wrote the as " This finding is consistent with the social displacement theory ".

Comment 22. There is no discussion of how much parent-child communication mediated the examined associations although this is reported in the results.

Response 22. Thank you for your comment. As in our response 20, we discussed the finding that in the comparison of other symptoms, the mediation effect of the communication was strongest on sleep difficulty symptom. And we discussed why.

Comment 23. There is no discussion of why confounders are not included. I guess this could be a bias in this study.

Response 23. Thank you for your suggestion. In this case, there were two demographic variables, sex and age. So in the discussed section, we discussed our limitation that we did not investigate if the mediation effect of communication with parents can vary across sex and age. Especially, regarding the variable "age", it would be better to investigate its effect by tracking adolescents longitudinally. 

Comment 24. Line 274: “making the cause-effect statement less strict in the current study”. What do the author mean with ‘less strict’? Cross sectional data cannot conclude a cause and effect relationships, however, it would be nice to include a discussion of whether there could be reverse causality and potential confounders.

Response 24. Thank you for your suggestion. And we agreed we should not use the word "less strict". So in the new manuscript, we noted "we could not make cause-effect statements in the current study". Moreover, we recommended the dynamic-data-based analysis in furthers' longitudinal researches to infer the cause-effect and reciprocal relationship between variables. Please find the added information "We recommend that researchers investigate the relationship between these variables through a longitudinal analysis in the future. Furthermore, a dynamic-data-based design is highly recommended because dynamic models emphasize the changes of variables in a development process, which helps us understand the cause-effect and reciprocal relationships between excessive electronic device use, parent-adolescent communication and psychological symptoms. For instance, as we discussed, there might be a longitudinal two-direction relationship between adolescents' excessive electronic device use and parent-adolescent communication. If adolescents spend excessive time online, they may have less communication with parents, and in turn, the lack of parent-adolescent communication may exacerbate adolescents' internet addiction.".

Comment 25. Line 278: It is stated that a dynamic-data-based design is recommended. However, it is unclear to me what this is.

Response 25. Thank you for your comment. In this new submitted version, as the statement in response 24, we explained more about the dynamic-data analysis, including the definition and the strengths .

Comment 26. Conclusion: The conclusion is not completely answering the aims. It could be improved by being more concise.

Response 26. Thank you for your valuable suggestion. We revised the conclusion section. First, we summarized the findings more concisely. And we address the implication subsequently. Please kindly find the re-edited conclusion.

Reviewer 2 Report

I greatly appreciate having the opportunity to review the manuscript, “Associations of Adolescents’ Excessive Electronic Device Use, Emotional Symptoms, Sleep Difficulty, and Communication with Parents: Two-wave Comparison in the Czech Republic.” I commend the authors for extending previous literature on adolescents’ electronic device use and its relation to parent-adolescent communication and psychological symptoms to a diverse population – Czech adolescents in 2006 and 2014. The findings add to the existing body of literature on children/adolescents’ media use and have solid findings. The paper, however, would benefit from heavy revision to better reflect the theoretical foundations in the Introduction and implications of the findings in Discussion. There was one major concern with methodology, pertaining to how the researchers measured parent-adolescent communication quality. Please find the detailed suggestions below (* means major concern).  

Introduction

·      [p.1, line 44] I strongly encourage the authors to edit the sentence, so the word “prove” is replaced with more nuanced vocabulary.

·      *[p.2, line 52-54] Given that the current study has a strong foundation on the social displacement theory (it was mentioned in the abstract), this section would benefit from elaborating more on the theory in a couple more sentences or a separate paragraph.

·      [p.2, line 57-58] The sentence, “In fact,…” needs edits for better readability.

·      *[p.2, line 51-64] This paragraph seems to list the studies related to adolescent-parent relation and adolescents’ psychological health. It would benefit from further explaining how these suggest a mediating relationship of adolescent-parent interaction and electronic device use and psychological symptoms.

·      *[p.2, line 80-82] I strongly suggest clarifying what “most vital symptom connected to excessive electronic device use” includes and adding a citation for this statement. Also, it would be helpful to elaborate more on the “most vital symptom” or psychological symptoms the authors are suggesting to relate to excessive electronic device use to better understand the potential hypothetical connection among the suggested variables.

Methods

·      [p.3, line 107] Please clarify “about every” for scale 1 on the rating. Every month? Week? Day?

·      [p.3, line 115] Please elaborate more what “for other purposes” mean. The examples are helpful but not sure which ones were not part of this.

·      *[p.3, Communication with Parents] The questions seem to gauge the adolescents’ perception of how easy they can talk to their parents rather than indirect behavioral measures of communication with parents. I am a bit concerned that the questions do not capture what the research question is asking (“adolescents’ communication with parents” as per the introduction last sentence). I am also unsure if it reflects the “quality of communication with parents.” Perhaps, the questions simply reflect the adolescents’ perception of their communication with parents or how approachable their parents are.

Results

·      The results and analyses seem solid.

Discussion

·      [p.7, line 212-213] For better readability, I suggest rephrasing (1) as “determine the psychological symptom most strongly related to adolescents’ excessive electronic device use.”

·      [p.7, line 235] The sentence “This finding consists of the social displacement theory…” sounds a bit awkward and may be beneficial for rephrasing.

·      [p.7, line 237-239] This sentence would also benefit from editing “who addict” to “who are addicted” for better readability.

·      *[p.7, line 246-248] This part is a bit confusing because it is written that the measure of electronic use was the participants’ reported time of electronic device use for other purposes. Maybe I misunderstood something here, but I am not sure how using electronic device for other purpose was the reason why the association between electronic device use and communication with parents was not found (since it was the primary dependent measure of electronic device use). It might be better to simply emphasize the difference in social media prevalence in 2006 and 2014.

·      *[p.8] The Discussion section would greatly benefit from elaborating more on the implications of the findings such as the development of guidelines for parents to supervise/manage media use of adolescents, directions for public policy, etc. This would warrant another paragraph.

Overall

·      [p.5, Figure 1 and Figure 2] Under “Group 1,” please make sure the words have spaces in between “sleepdifficulty” and “feellow.”

·      I strongly suggest the authors to keep the words consistent across the paper – some parts of the paper use “parent-adolescent communication” while other places use “adolescent-parent communication.”

Overall, I applaud the authors’ effort at extending the literature to Czech adolescents’ media use and its implications on parent-adolescent communication and psychopathological symptoms. With the concerns addressed, I believe the paper could add to the growing literature on how media use affects adolescents and their family. I sincerely hope the authors will find my review helpful.

Author Response

Dear reviewer, thank you very much for your valuable comments and suggestions! Based on all your comments, we would like to respond. Please kindly find the responses below. Thanks again!

Comment 1. Introduction

  • [p.1, line 44] I strongly encourage the authors to edit the sentence, so the word “prove” is replaced with more nuanced vocabulary.

Response 1. Thank you for your suggestion. We replace the word "proved" with "certified" .

Comment 2·      *[p.2, line 52-54] Given that the current study has a strong foundation on the social displacement theory (it was mentioned in the abstract), this section would benefit from elaborating more on the theory in a couple more sentences or a separate paragraph.

Response 2. Thank you very much for your suggestion. In the new revised version, we elaborated the social displacement theory more. Please find the added text "Social displacement theory initially marked the effect of media on interpersonal relationships. It suggested that time spent in media replaces the time spent in face-to-face interaction with family and friends, who are highly correlated to individuals' wellbeing. Before the personal computer (PC) era, TV was the main contributor to social displacement. In the PC-era and current post-PC era, the internet and social media are seen as the decisive reasons for social displacement respectively [11,12].". Thank you.

  • Comment 3. [p.2, line 57-58] The sentence, “In fact,…” needs edits for better readability.
  • Response 3. Thank you for your suggestion. This sentence was re-edited like "In fact, it was found that adolescents' intensive social media use is indirectly associated with depressive symptoms through perceived parental support, which meant during the parent-adolescent interaction, adolescents' subjective feelings of parental support mediate the relationship between adolescents' too much social media use and their depression" in the new manuscript.

 Comment 4. *[p.2, line 51-64] This paragraph seems to list the studies related to adolescent-parent relation and adolescents’ psychological health. It would benefit from further explaining how these suggest a mediating relationship of adolescent-parent interaction and electronic device use and psychological symptoms.

Response 4. Thank you for your comment. Actually, this paragraph was mainly discussing the mediator role of adolescent-parent interaction in the relationship between electronic device use and psychological symptoms. Maybe we provided too many details in the old version and they may distract readers from the key point, thus, we added the matched summary for each listed study. Moreover, we emphasized the mediator role of parent-adolescent interaction has been more and more important after stepping into today's post-PC era. Please find the revised paragraph in the new manuscript. Thank you so much.  

.    Comment 5. *[p.2, line 80-82] I strongly suggest clarifying what “most vital symptom connected to excessive electronic device use” includes and adding a citation for this statement. Also, it would be helpful to elaborate more on the “most vital symptom” or psychological symptoms the authors are suggesting to relate to excessive electronic device use to better understand the potential hypothetical connection among the suggested variables.

Response 5. Thank you very much for your comment and suggestion. In fact, our aim was to compare the strengthens of correlations between each psychological symptom and excessive electronic device us and find out the symptom connected to excessive electronic device use most strongly in 2006 and 2014. So in our new version, we added an additional paragraph (please see the second last paragraph in introduction section) to explain there was not a study to investigate the associations between excessive electronic device use and various of psychological symptom. And, in the last paragraph, we explained the definition of "most vital symptom". Also, we additionally introduced the main categories of psychological symptoms, and we provided Gariepy and his colleagues' study as the reference. Please kindly find these changes in the new manuscript. Thanks.

Comment 6.Methods

  • [p.3, line 107] Please clarify “about every” for scale 1 on the rating. Every month? Week? Day?

Response 6. Thank you for pointing out our mistake. The item response was revised as "about every day".

  • Comment 7. [p.3, line 115] Please elaborate more what “for other purposes” mean. The examples are helpful but not sure which ones were not part of this.

Response 7. Thank you very much for your comment. In the new manuscript, we presented the original items in 2006 and 2014. Actually, the item in 2014 was "how many hours a day, in your free time, do you usually spend using electronic devices such as computers, tablets (like iPad) or smart phones for other purposes, for example, homework, emailing, tweeting, Facebook, chatting, surfing the internet for weekdays and weekend". Thus, according to the item in HBSC questionnaire, the "other purposes" refers to the purpose beyond just practicing using electronic device itself, but for other purposes, like using social media, emailing, and etc.

  • Comment 8. *[p.3, Communication with Parents] The questions seem to gauge the adolescents’ perception of how easy they can talk to their parents rather than indirect behavioral measures of communication with parents. I am a bit concerned that the questions do not capture what the research question is asking (“adolescents’ communication with parents” as per the introduction last sentence). I am also unsure if it reflects the “quality of communication with parents.” Perhaps, the questions simply reflect the adolescents’ perception of their communication with parents or how approachable their parents are.

Response 8. Thank you for your question. Please allow me to explain it here and in the main text. The original item was "how easy is it for you to talk to your father about things that really bother you". And the responses were 1 ("very easy"), 2 ("easy"), 3("difficult"), 4("very difficult"), and 5 ("do not have or see this person"). Thus, mainly, this item was aimed to investigate the parent-adolescent communication quality, instead of investigating how approachable their parents are. The supportive talks between parents and adolescents about daily problems are highly related to adolescents' wellbeing [24]. The easy talk about life problems reflects good communication quality, and the difficult talk reflects poor communication quality. The communication quality is considered the worst if parents totally refuse to talk about their children's life problems. Please also find these explanation in our new manuscript. Thank you a lot!

Comment 9. Results

  • The results and analyses seem solid.

Response 9. Thank you very much for your positive feedback.

Comment 10.Discussion

  • [p.7, line 212-213] For better readability, I suggest rephrasing (1) as “determine the psychological symptom most strongly related to adolescents’ excessive electronic device use.”

Response 10. Thank you very much for your suggestion. We revised the sentence as " The primary purposes of the study were (1) to determine the psychological symptom most strongly related to adolescents’ excessive electronic device use ; "

  • Comment 11. [p.7, line 235] The sentence “This finding consists of the social displacement theory…” sounds a bit awkward and may be beneficial for rephrasing.

Response 11. Thanks for pointing out it. We revised the sentences as " This finding is consistent with the social displacement theory". Please find the change in the new manuscript.

  • Comment 12. [p.7, line 237-239] This sentence would also benefit from editing “who addict” to “who are addicted” for better readability.

Response 12. Thank you for your suggestion. We replaced "who addict" with "who are addicted" in the new version.

  • Comment 13 *[p.7, line 246-248] This part is a bit confusing because it is written that the measure of electronic use was the participants’ reported time of electronic device use for other purposes. Maybe I misunderstood something here, but I am not sure how using electronic device for other purpose was the reason why the association between electronic device use and communication with parents was not found (since it was the primary dependent measure of electronic device use). It might be better to simply emphasize the difference in social media prevalence in 2006 and 2014.

Response 13. Thank you for your comment. First, in response to your comment 7, we presented the original item for the measurement. The "other purposes" in the questionnaire item refer to using electronic device beyond the purpose of practicing using the device, for instance, using social media and emailing. We are sorry for we used the same word "other purposes" in this paragraph, which made you misunderstand. So we re-edited the sentences : "The reason might be that, in 2006 adolescents mainly used electronic device for purposes beyond social media, such as searching for information for homework, reading, and emailing, rather than connecting to social media." Next, we noted that, after stepping into the post-PC (personal computer) era, which started in the late 2000s and early 2010s, the trend of adolescents' daily social media usage through smart phone has increased significantly. Excessive social media use decreases the communication with parents in the offline world. We are sorry that we cannot find the exact prevalence of social media use in 2006 and 2014. But as mentioned, we cited an article to prove the increasing trend of social media use among adolescents.

  • Comment 14. *[p.8] The Discussion section would greatly benefit from elaborating more on the implications of the findings such as the development of guidelines for parents to supervise/manage media use of adolescents, directions for public policy, etc. This would warrant another paragraph.

Response 14. Thank you for your suggestion very much. In the new manuscript, we listed some tips for parents to manage adolescents' social media use. First, parents need to communicate with their children frequently. Especially, parents should encourage children to talk about problems that they meet. Second, it is necessary for parents to avoid accelerating parent-adolescent conflict. Third, it is a good idea for parents to discuss with the children together about excessive electronic device usage problem and find out the deep reason behind it. Please find the revisions in the new version.

Overall

  • Comment 15.  [p.5, Figure 1 and Figure 2] Under “Group 1,” please make sure the words have spaces in between “sleepdifficulty” and “feellow.”

Response 15. Thank you for your comment. We corrected the figures. Please kindly find the figures in the new manuscript.

  • Comment 16. I strongly suggest the authors to keep the words consistent across the paper – some parts of the paper use “parent-adolescent communication” while other places use “adolescent-parent communication.”

Response 16. Thank you very much for your suggestion. We keep the word "parent-adolescent communication" consistent across the whole paper in the new manuscript.